# Orbital selective commensurate modulations of the local density of states in $ScV_6Sn_6$ probed by nuclear spins

Robin Guehne [1] ✉, Jonathan Noky[1], Changjiang Yi[1], Chandra Shekhar [1], Maia G. Vergniory[1,2], Michael Baenitz [1] & Claudia Felser[1]

The kagome network is a unique platform that harbors a diversity of special electronic states due to its inherent band structure features comprising Dirac cones, van Hove singularities, and flat bands. Some kagome-based metals have recently been found to exhibit favorable properties, including superconductivity, charge order, and signatures of an anomalous Hall effect. The kagome system $ScV_6Sn_6$ is a promising candidate for studying the emergence of an unconventional charge order and accompanying effects. We use $^{51}V$ nuclear magnetic resonance to explore the local properties of the charge ordered phase in single crystalline $ScV_6Sn_6$, aided by density functional theory. We show the local charge symmetry of V to reflect a commensurate modulation with $\mathbf{q} = (\frac{1}{3}, \frac{1}{3}, \frac{1}{3})$, the density of states to drop by about a factor of $\sqrt{2}$ during the phase transition, and an unusual orientation dependent change in the shift splitting symmetry to reveal orbital selective modulations of the local density of states.

Covalent two-dimensional metals with triangular motifs and confined electronic states are a rich playground in modern solid states physics. Among the non-magnetic materials, systems that are based on kagome layers attract increasing attention as they naturally feature Dirac cones, van Hove singularities, and flat bands in their electronic band structure, that may further chemically be tuned to the vicinity of the Fermi level[1]. Lately, the V based kagome metal $CsV_3Sb_5$ appeared in the spotlight as it undergoes a charge density wave (CDW) phase transition at ~94 K before it become superconducting below about 2 K, providing the prospect to learn more about the interplay of both phenomena[2]. Evidence for a manipulable chiral transport as well as an anomalous Hall effect (AHE) in the CDW regime further add to the rather diverse list of special electronics of this kagome metal[3,4].

Recently, the closely related bilayer kagome material $ScV_6Sn_6$ was reported to undergo a CDW transition at about 92 K, including evidence for topologically non-trivial bands and signatures of an anomalous Hall effect, while superconductivity was not observed at ambient pressure[5-25]. In particular the unconventional nature of the CDW phase with a 3-dimensional wavevector $\mathbf{q}_3 = (\frac{1}{3}, \frac{1}{3}, \frac{1}{3})$ gives rise to an ongoing debate about the formation mechanism of the charge order[6-15,17-22]. Some experiments and theory further suggest a high-temperature short-range CDW with $\mathbf{q}_2 = (\frac{1}{3}, \frac{1}{3}, \frac{1}{2})$ that is unstable at low temperatures[10-12,14,22].

The complexity of the CDW as well as numerous experimental reports of unusual phenomena led us to employ nuclear magnetic resonance (NMR) to characterize the CDW phase of single crystalline $ScV_6Sn_6$ and to search for signatures of an unusual magnetism. Given its high sensitivity for chemical and electronic properties, as well as the atomic resolution, NMR has a long tradition as a method to investigate charge-ordered systems, as e.g., in the case of the famous dichalcogenides[26]. In the context of cuprates, NMR is used to deepen the understanding of the interplay between charge order and superconductivity[27,28]. Similarly, numerous NMR studies of $CsV_3Sb_5$ explored the microscopic structure of the kagome lattice in the CDW phase, with a strong evidence for a tri-hexagonal configuration of the V atoms[1,29-35].

With this paper we present a comprehensive single crystal $^{51}V$ NMR study of the kagome metal $ScV_6Sn_6$ aided by density functional

[1]Max Planck Institute for Chemical Physics of Solids, 01187 Dresden, Germany. [2]Donostia International Physics Center, 20018 Donostia - San Sebastian, Spain. ✉e-mail: robin.guehne@cpfs.mpg.de

theory (DFT). We explore the dynamic properties of the local magnetic field during the CDW phase transition between 96 and about 80 K, and determine a drop in the density of states (DOS) by about a factor of $\sqrt{2}$ (1.42(2)). The CDW phase is characterized as a commensurate, three-fold modulation of the local V charge symmetry in accordance with the reported reconstruction of the unit cell with periodicity $\sqrt{3} \times \sqrt{3} \times 3$. Our NMR shift data further reveal a peculiar phase shift $\Delta\varphi = \frac{\pi}{2}$ of the sinusoidal modulation of the local magnetic field between the in-plane and the out-of-plane field orientation. This observation can be explained by orbital selective modulations of the local DOS resulting in a distinct wave vector for the V $3d_{z^2}$ orbitals with $\mathbf{q}_{z^2} = (\frac{1}{3}, \frac{1}{3}, \frac{2}{3})$. Finally, with NMR we neither find direct evidence of an unusual magnetism as related to time-reversal symmetry breaking and an AHE, nor of an additional, high temperature charge modulation with $\mathbf{q}_2 = (\frac{1}{3}, \frac{1}{3}, \frac{1}{2})$.

## Results

### Charge density waves and nuclear magnetic resonance

In the 1960s, Overhauser considered a sinusoidal modulation of the local spin and charge density in metals which is translated into characteristic spectral changes of the corresponding NMR signal through the variation in the NMR shift or quadrupole interaction (as will be discussed in more detail below)[36]. Consider a one-dimensional (1D) sinusoidal modulation of the local electronic density,

$$\rho = \rho_0 + \delta\rho \cos(q \cdot x + \varphi). \tag{1}$$

Here, $\rho_0$ denotes the average or un-modulated charge in the CDW regime, $\delta\rho$ the amplitude, $q$ the wave vector in units of $\frac{2\pi}{a}$, $x$ the real-space position, and $\varphi$ the phase of the CDW. That is, different to most other methods, NMR can also determine the phase $\varphi$ of a charge modulation which can be an important detail in terms of the dynamics of CDWs[37]. A variety of commensurate CDW scenarios beside the corresponding NMR signals are illustrated in Fig. 1. We assume a periodic lattice of atoms with lattice parameter $a$ and a uniform local electron density $\rho_0$ (panel (a)). The NMR signal appears at a $K_0$ that corresponds to the uniform local charge. In panels (b)–(d) of Fig. 1 we show three different CDWs and their NMR resonance patterns. The charge density variation is represented by the size of the circles. The

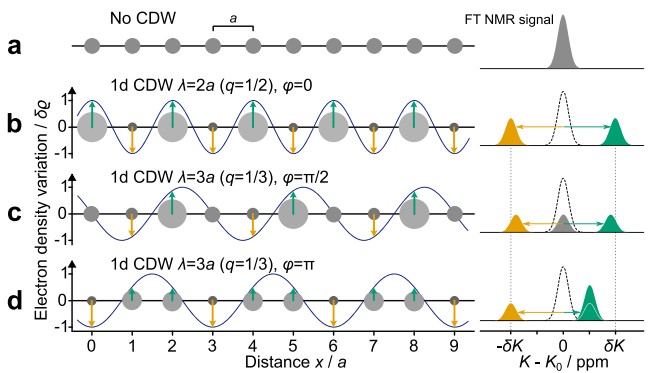

**Fig. 1 | Commensurate charge density waves and nuclear magnetic resonance.** **a** A chain of equidistant atoms with evenly distributed electron density (gray circles) yields a single nuclear magnetic resonance (NMR) signal at $K_0$ (including intrinsic broadening). **b** For a charge density wave (CDW) of amplitude $\delta\rho$, periodicity $q = \frac{1}{2}$, and phase $\varphi = 0$, local charge density peaks and valleys alternate between neighboring atoms, yielding a symmetric splitting into two resonance lines at $\pm\delta K \propto \delta\rho$. The signal intensity is proportional to the number of nuclei with the same local environment. **c** A CDW with $q = \frac{1}{3}$ yields 3 equally intense resonance lines. For $\varphi = \frac{\pi}{2}$, two of them are symmetrically shifted to higher and lower frequencies, while the third one remains un-shifted (no extra charge). **d** For the special case of $\varphi = \pi$, two of the 3 lines overlap at $\frac{\delta K}{2}$, while the third one is shifted by $\delta K$ in the opposite direction.

NMR line splits with a maximum of $2\delta K \propto \delta\rho$. Here, NMR shifts to higher and lower frequencies display the gain or loss of electron density for the respective crystal position. In general, the sign of $\delta K$ does not allow a conclusion on whether the corresponding nucleus sits in a valley or on a peak of the CDW, as it depends on the type of hyperfine interaction[38]. For $q = \frac{1}{2}$, the resonance line always splits symmetrically, independent of the CDW phase. Contrastingly, for $q = \frac{1}{3}$, as in case of $^{51}$V NMR of ScV$_6$Sn$_6$, the relative position of the three resonances depends on $\varphi$ (cf. panels (c, d)). This simple model is easily extended to incommensurate CDWs that give rise to broadened NMR spectra with a characteristic double-peak structure at $\pm\delta K$[28,36].

### Structural phases of ScV$_6$Sn$_6$ and local vanadium charge symmetry

The high-temperature phase of ScV$_6$Sn$_6$ has P6/mmm symmetry. The unit cell consists of 13 atoms, where chemically equivalent vanadium atoms form the kagome planes, cf. Fig. 2a. Below about 96 K (transition temperature from current study), ScV$_6$Sn$_6$ undergoes a structural phase transition yielding a change in the Sc and Sn positions along the crystal $c$-axis, while the V lattice remains almost unperturbed[6]. In panel (b) of Fig. 2, we show the relevant displacements in terms of Sn-Sc-Sn chains (gray-purple-gray) containing two kagome layers (red). Different to the symmetric arrangement characteristic for the high $T$ phase, the low $T$ phase features 3 Sn-Sc-Sn chains that vary in length and in relative distance to the V kagome plane. From the viewpoint of the V kagome lattice, we identify 3 corresponding Sn positions (1, 2, 3) that hover just above (below) the hexagonal V voids, creating 3 non-equivalent V crystal sites (V1, V2, V3), each of them neighbored by two different Sn sites as shown in the lower panel of Fig. 2b.

The structural phases feature different local V environments. NMR allows to differentiate them via the electric quadrupole interaction, if such environments have non-cubic symmetry and the nuclear spin satisfies $I > 1/2$ ($^{51}I = \frac{7}{2}$). In first order approximation, the quadrupole interaction can be written as

$$\mathcal{H}_Q = \frac{3I_z^2 - I(I+1)}{4I(2I-1)} eQV_{ZZ} \times \frac{1}{2}\left(3\cos^2\beta - 1 + \eta\sin^2\beta\cos 2\alpha\right), \tag{2}$$

where $eQ$ is the nuclear quadrupole moment and $V_{ZZ}$ is the principle value of the electric field gradient (EFG). The EFG is derived from the charge distribution at the nucleus' position, and is thus tightly connected to the above-mentioned local environment. The EFG is represented as a traceless ($V_{ZZ} + V_{YY} + V_{XX} = 0$), second-rank tensor. It is conveniently expressed in terms of its size ($|V_{ZZ}|$) and shape given by the asymmetry parameter $\eta = \frac{V_{XX} - V_{YY}}{V_{ZZ}}$ ($|V_{ZZ}| \geq |V_{YY}| \geq |V_{XX}|$). The Euler angles $\alpha$ and $\beta$ determine the relative orientation of the EFG's principle axis system with respect to the external magnetic field (typically laboratory $z$-axis), and thus with respect to the sample's crystal lattice.

From (2) it follows that the V nuclear spin system splits into $2I = 7$ equidistant resonance lines (note, second order shifts amount to about 1 kHz in maximum), 1 central transition (CT) and 3 pairs of satellites, as shown in the top spectrum of Fig. 3a. The anisotropy of the interaction ($\alpha$ and $\beta$) is transferred to the apparent quadrupolar line splitting $\tilde{\nu}_Q$, i.e., the splitting depends on the sample orientation in the external field, as

$$\tilde{\nu}_Q = \frac{3eQV_{ZZ}}{4I(2I-1)}\left(3\cos^2\beta - 1 + \eta\sin^2\beta\cos 2\alpha\right), \tag{3}$$

with the quadrupole splitting frequency defined as

$$\nu_Q = \frac{3eQV_{ZZ}}{2I(2I-1)}. \tag{4}$$

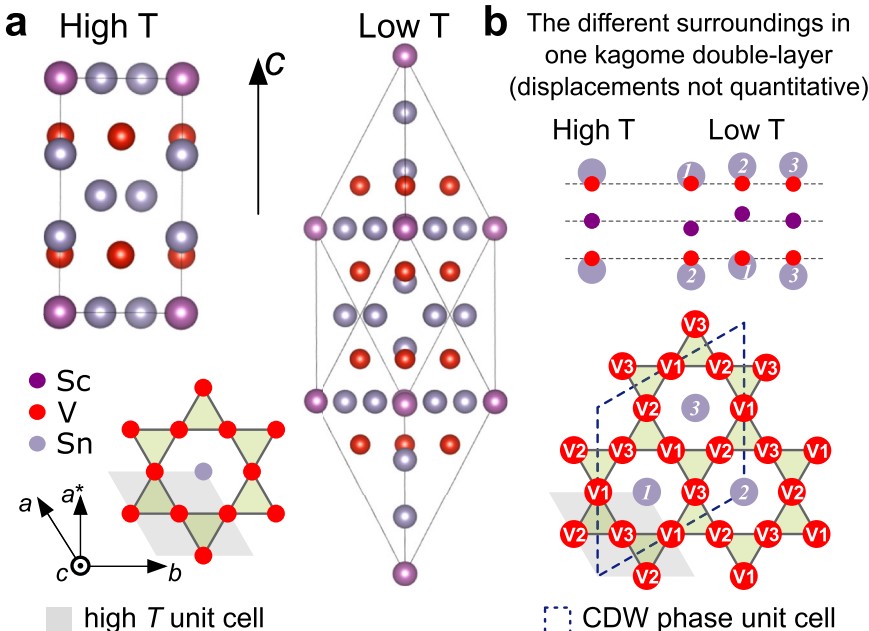

**Fig. 2 | Crystal structures of ScV$_6$Sn$_6$. a** On the left, side view of the high temperature unit cell of ScV$_6$Sn$_6$, and below, top view of the V kagome lattice (red). In the charge density wave (CDW) phase, the relative position of the Sn-Sc-Sn chains (gray and purple) is altered along the crystal *c*-axis, yielding 3 different Sc-Sn-Sc units as depicted in panel (**b**) containing one V kagome double layer each. This expands the unit cell by a factor of three. The displacement affects the distance of the kagome-close Sn atoms (*1,2,3*), creating 3 distinct V crystal sites (V1, V2, V3).

By evaluating $\tilde{\nu}_Q$ for various crystal orientations, size ($|V_{ZZ}|$), shape ($\eta$), and orientation of the EFG can be determined. In Fig. 3b we show an example of this procedure for the high-temperature phase (a detailed account is provided in the supplementary information), which corresponds to $\tilde{\nu}_Q(\beta, \alpha = 0)$, i.e., orientation-dependent measurements under crystal rotation about the *c*-axis (Supplementary Fig. 3). We find that $V_{ZZ}$ points along the crystal $a^*$-axis with $\nu_Q = 475(2)$ kHz and $\eta = 0.83(1)$. The three identical data sets (triangles, squares, and circles) shifted by ±60° recover the sixfold symmetry of the kagome lattice (insets) and prove the equivalency of the three corners of each V kagome triangle in terms of the local vanadium symmetry. To visualize the V EFGs in the kagome lattice, we display them as ellipsoids formed by $|V_{ZZ}|$, $|V_{YY}|$, and $|V_{XX}|$, as shown in panels (c) and (d) of Fig. 3.

Upon cooling, between 96 and about 80 K, the phase transition takes place and the low-temperature structure replaces the high *T* one. From the NMR point of view (Fig. 3a lower spectrum), the changes are most prominently visible in a splitting of the low-frequency satellites (find a detailed account of the temperature-dependent spectra in Supplementary Fig. 1). The single quadrupole pattern observed for high temperature is replaced by 3 sets of quadrupole spectra. In panel (e) of Fig. 3 we show the results of the same orientation-dependent measurements as in panel (b), now performed at 80 K (Supplementary Fig. 5). The change to a threefold quadrupole interaction is consistently found in any possible direction. This proves that the local vanadium environments have changed, while the relative orientation of the 3 EFGs is the same as at high temperatures. The arrangement of the EFGs is illustrated in panel (f). When viewed along the crystal *a* or *b* axis, panel g, the 3 alternating EFGs create a commensurate 1D sinusoidal variation with $q = \frac{1}{3}$, $\varphi = \frac{\pi}{4}$, and an amplitude of about 20 kHz around the average quadrupole splitting of 483.7 kHz. Detailed DFT calculations based on the structural models provided by Arachchige et al.[6] as implemented in VASP[39,40] confirm the results shown in Fig. 3, cf. Table 1. Note that with the experiment we cannot determine the sign of $V_{ZZ}$ since the quadrupole splitting is symmetric.

## Charge density wave phase transition

In Fig. 4a we show the center line of the quadrupole spectra from Fig. 3a, for temperatures between 100 K and 20 K. For the sake of clarity, we have removed the (very small) temperature dependence of the NMR shift. At 96 K, first indications of an additional signal (blue) appear -15 kHz (-150 ppm) next to the initial resonance line (gray). As this shoulder grows in intensity with decreasing temperature, the initial line decreases, until, for -80 K and below, the overall shape of the doublet remains unchanged.

In order to track the volume fractions of the two competing phases during the phase transition, we used Gaussian lines to fit the two resonances in panel (a) of Fig. 4 and extracted the corresponding signal intensities. The results are shown panel (b). The gray and blue data points denote the respective signal intensities of panel (a) (for an analysis of the total signal intensity, the reader is referred to Supplementary Fig. 9). The individual values are normalized with respect to their sum for each temperature. In the course of cooling, the low temperature resonance line (blue) grows to about $\frac{2}{3}$ of the total intensity, leaving about $\frac{1}{3}$ (gray) to what appears to be the initial high temperature line. With the colored dashed lines in Fig. 4b we show the true changes in the relative intensities, concluded from the corresponding changes in the quadrupole splittings. The initial high-temperature CT disappears entirely between 96 and about 80 K (black dashed line). It is replaced by three resonances (purple, orange, and light blue) with intensity ratio 1:1:1 that represent the 3 CTs of the low-temperature quadrupole patterns identified in the former section. Two of the 3 resonances (orange and light blue) form the dark blue peak that appears during the phase transition in panel (a).

In Fig. 4c, d we show the temperature dependence of the NMR shift for the out-of-plane orientation of the magnetic field ($c\|B_0$) and for the in-plane orientation ($a^*\|B_0$). In the latter case, a reliable determination of the shift values during the phase transition was not possible because of the overlap of high and low-temperature resonance lines. For 80 K and below, the shift was determined via the satellite transitions (Supplementary Fig. 6). We assigned the shift values between $c\|B_0$ and $a^*\|B_0$ as shown by the colors based on the DFT

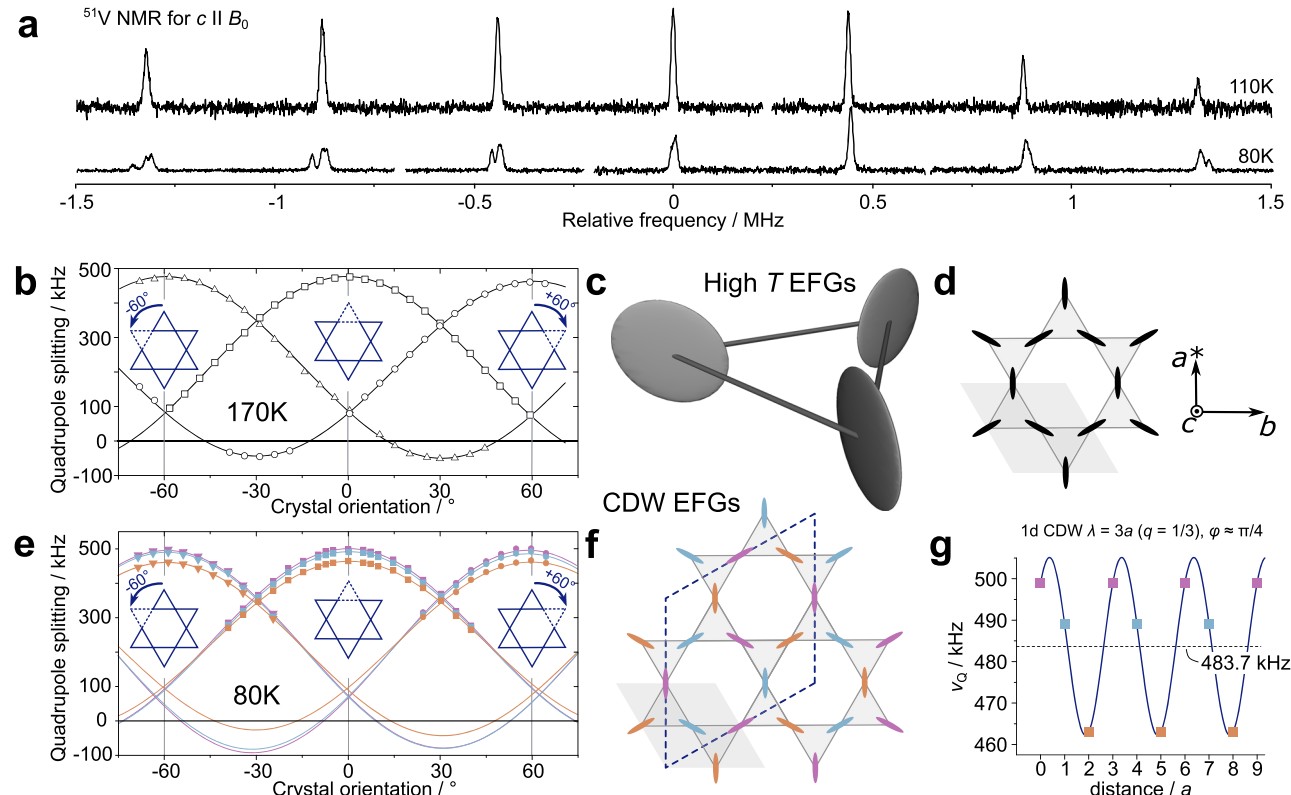

**Fig. 3 | Local vanadium charge symmetry for high and low temperatures. (a)** The Fourier transform of the $^{51}$V nuclear magnetic resonance (NMR) spectra (from selective spin echoes) for 8.73 T and $c\|B_0$. The spectra were obtained at 110 and 80 K, confronting the spectral differences between the high and the low temperature phase. The central frequency has been subtracted for clarity. **(b)** High temperature orientation dependent quadrupole splittings for rotations about the crystal $c$-axis (magnetic field in the $ab$-plane) reveal the sixfold rotation symmetry of the kagome lattice (Supplementary Fig. 3). Perspective view **(c)** and top view **(d)** of the equivalent electric field gradients (EFGs) at V sites in the kagome lattice. **(e)** Low temperature orientation dependent quadrupole splittings for in-plane

rotations confirm the three sets of quadrupole patterns already observed in panel **(a)**, while the sixfold rotation symmetry from the high $T$ experiments is retained (Supplementary Fig. 5). The sixfold rotational symmetry reflects the threefold symmetry of a kagome layer in the CDW regime and that the two neighboring planes of the kagome double layers are mirror images of each other. **(f)** Thus, for low temperatures, the three corners of each V triangle become chemically non-equivalent while the EFGs keep the orientation of the high $T$ phase. **(g)** The commensurate variation of the EFGs along the crystal $a$ or $b$ axis, can be viewed as a 1D sinusoidal modulation with $q = \frac{1}{3}$ and $\varphi \approx \frac{\pi}{4}$.

## Table 1 | Electric field gradients at vanadium sites

| | High $T$ | | Low $T$ | | | | | |
|---|---|---|---|---|---|---|---|---|
| | NMR | DFT | NMR1 | DFT1 | NMR2 | DFT2 | NMR3 | DFT3 |
| $v_Q$ [kHz] | 475(2) | | 463(2) | | 489(2) | | 499(2) | |
| $V_{ZZ}$[$10^{20}$V/m$^2$] | 52.9(2) | −60.812 | 51.6(2) | −59.679 | 54.4(2) | −61.861 | 55.6(2) | −62.617 |
| $\eta$ | 0.83(1) | 0.809 | 0.91(4) | 0.841 | 0.81(4) | 0.782 | 0.78(3) | 0.787 |

The electric quadrupole splitting frequency $v_Q$, the principle axis of the electric field gradient (EFG) $V_{ZZ}$ from (3) (for the experiment, only the absolute value can be obtained, the sign cannot be accessed), and the asymmetry parameter $\eta$ as obtained from NMR and density functional theory (DFT) at V positions above (High $T$) and below (Low $T$) the charge density wave phase transition.

calculations. The assignment cannot be proven with experiment because the superposition of many resonance lines for orientation-dependent measurements prevents a reliable tracing of individual signals. This, however, has no effect on the conclusions of the present investigation.

The V shift in ScV$_6$Sn$_6$ is with more than about 0.6% and 0.8% (6000–8000 ppm) very large, strongly anisotropic, and only very weakly dependent on temperature (2–4% of the total shift). From detailed orientation-dependent measurements (Supplementary Figs. 2–4) it can be seen that the high-temperature shift tensor is axially symmetric with the main axis pointing toward the center of the V triangle (along the $a^*$ direction) similar to the EFG. Although it is very difficult to prove experimentally, it is reasonable to assume that the CDW phase inherits the axial shift anisotropy. Isotropic and axial shift

components for the full range of temperatures are depicted in Supplementary Fig. 7.

In panel (a) of Fig. 5 we show the spin-lattice relaxation rate of V nuclei in the two different samples, for two orientations ($c\|B_0$ and $a^*\|B_0$), and for a wide range of temperatures. After continuously decreasing between room temperature and about 100 K, $T_1^{-1}$ drops abruptly in the region of the phase transition by about a factor of 2 (1.95). Between 80 and 4 K, the relaxation rate is proportional to the temperature. Measurements for $a^*\|B_0$ (yellow circles) prove the spin-lattice relaxation to be isotropic (see also Supplementary Fig. 11). In the low-temperature phase, the data points represent the averaged relaxation of the 3 components. In the inset we present the vanadium DOS for both phases obtained from DFT. The high-temperature DOS are denoted by the solid line, the CDW phase DOS by the dashed line.

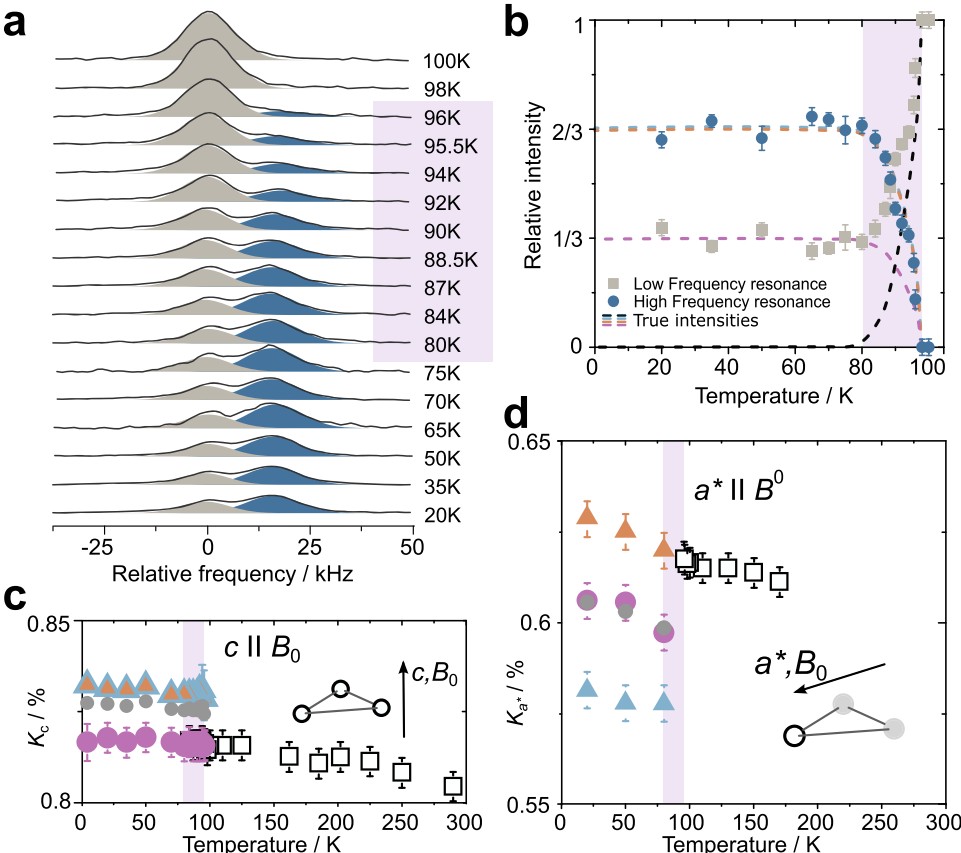

**Fig. 4 | Charge density wave phase transition and the local magnetic field. a** The central region of the $c\|B_0$ spectrum as function of temperature at 8.73 T (Fourier transform of selective spin echoes). Highlighted is the temperature range of the phase transition. Gaussian lines (gray and blue) were used to fit the spectra and to extract the intensity changes. **b** The area of the Gaussians (proportional to signal intensity) from panel (**a**) as a function of temperature. Error bars were obtained from the Gaussian fits and reflect the varying signal-to-noise ratios of the individual measurements. Dashed lines denote true relative intensities. **c, d** The temperature dependence of the NMR shift for $c\|B_0$ and $a^*\|B_0$. Error bars are related to the finite line broadening of the resonances. Note that "$a^*\|B_0$" holds only for 1 of the 3 V nuclei per V triangle as shown in the inset. Gray data points denote the average of the two and three low temperature shift values.

At the Fermi level, the difference between high and low temperature DOS is 1.44, i.e., very close to $\sim\sqrt{2}$ as expected from the observed changes in the relaxation (cf. discussion). The temperature dependence of $(T_1T)^{-1}$, as shown in panel (b), yields a constant value for temperatures below 80 K and a pronounced step during the phase transition. For temperatures above $T_{CDW}$ and $\leq 150$ K, $(T_1T)^{-1}$ appears to be rather constant as well, while it clearly deviates from such a behavior for even higher temperatures. This progressive increase implies a gain in available states as, e.g., in the case of thermal excitation across an energy gap. An activation type of fit (blue line) gives a gap of -160 meV. In panel (c) we show the corresponding Arrhenius plot for the high temperature relaxation rate (above 150 K).

## Discussion

The temperature dependent $^{51}$V spectra of ScV$_6$Sn$_6$ as shown in Fig. 3 (Supplementary Fig. 1) and 4 provide consistent evidence of a first order phase transition between 96 and about 80 K in good agreement with literature[16]. The low $T$ phase is similarly homogeneous as the high $T$ one, with no additional broadening, neither in the NMR shift nor in the quadrupole interaction, proving any observed changes in the static NMR at low temperatures are perfectly commensurate and sufficiently long range such that domain effects are negligible. Hence, our observations do not support the presence of an additional short-range intermediate charge order[10–12,14,22]. It should be noted, however, that the reported CDW phase chiefly considers

displacements of Sn along the $c$-direction, which may have only a very weak effect on the V NMR. Possible consequences are unusual EFG components (splitting or broadening) or an unusual contribution to the spin-lattice relaxation due to the corresponding fluctuations that vanish during the CDW phase transition[12,14]. None of these signatures were observed.

Our results also do not provide any direct evidence of an unusual magnetism as related to TRS breaking and an anomalous Hall effect. The $c\|B_0$-splitting in units of ppm as well as the line shape, i.e., the relative intensity of this double-peak structure, are independent of the applied magnetic field, and thus point at different local spin densities as expected for a CDW (cf. Supplementary Fig. 10). Similarly, total signal intensity as well as excitation conditions of the NMR experiments do not indicate any effect from magnetism, as there are no unusual losses or enhancements. That is, any additional effect must be well below the observed linewidth of our resonance lines which is in the order of 10 kHz (-100 ppm or -1 mT at 8.73 T external field). This implies, for example, that a magnetic moment of $<10^{-2}\mu_B$ localized at the center of V hexagons (1.75 Å distance to V[6]) remains undetectable with our setup, while, perhaps, be sensed by muon-spin rotation[25].

For a metallic system like ScV$_6$Sn$_6$, the spin-relaxation is typically governed by magnetic relaxation, i.e., by fluctuations of the local magnetic field arising from conduction electrons (Supplementary discussion and Supplementary Fig. 11). The temperature dependence of the relaxation rate $1/T_1$ should then be proportional to the square of

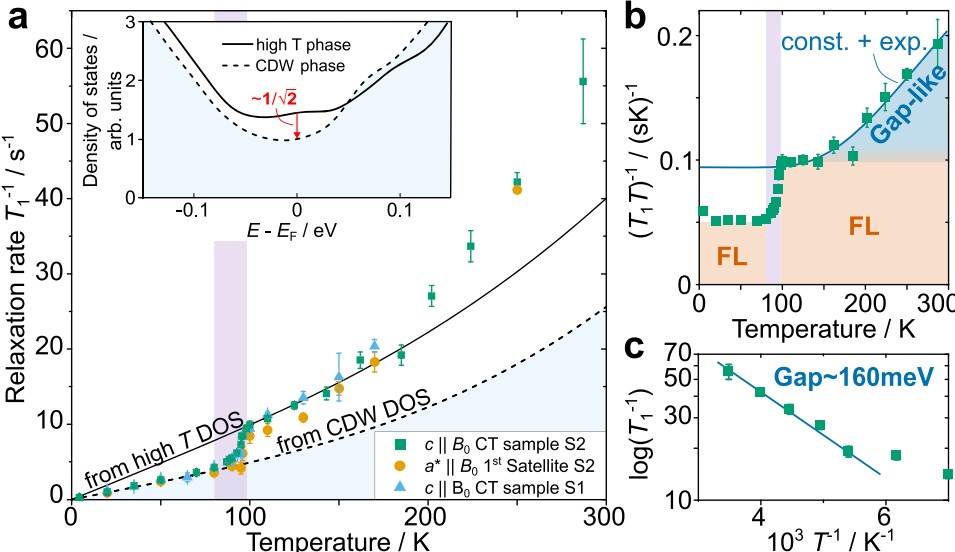

**Fig. 5 | Charge density wave phase transition and fluctuations of the local magnetic field. a** The spin-lattice relaxation rate $T_1^{-1}$ as a function of temperature obtained from the central transition (CT) for ($c\|B_0$) for sample S1 and S2, and from the first low frequency satellite for $a^*\|B_0$. Error bars were obtained from the fitting procedure as discussed in Supplementary Note 8. The pink-shaded temperature range denotes the phase transition. The inset shows the calculated density of states (DOS) for V states at high (solid) and low (dashed line) temperatures. At $E_F$, the DOS

drops by about $\sqrt{2}$ due to the charge density wave (CDW) phase transition. **b** $(T_1T)^{-1}$ as a function of temperature. Pink rectangle denotes the CDW phase transition as in panel (**a**). Constant $(T_1T)^{-1}$ represents Fermi liquid (FL) behavior. The blue solid line represents a fit for the high temperature phase data using $\left(const. + \frac{\bar{T}}{T} \times \exp\left(-\frac{\Delta E/2}{k_BT}\right)\right)$ yielding $\bar{T} \sim 730$ K and $\Delta E \sim 160$ meV. Panel **c** shows the corresponding Arrhenius plot for the high temperature region above 150 K.

the DOS (Korringa relation). We evaluated the following integral,

$$\frac{1}{T_1} \propto A_{hf}^2 \int f(E - \mu_c)[1 - f(E - \mu_c)]D(E)^2 dE , \qquad (5)$$

where $A_{hf}$ represents the hyperfine interaction that we assume to be temperature independent, $f(E)$ denotes the Fermi function, $\mu_c$ the temperature dependent chemical potential, and $D(E)$ the energy dependence of the DOS for the high and the low temperature phase from the inset of Fig. 5a. The results are shown by the dashed and the solid black line in Fig. 5a after appropriate rescaling. We find a very good agreement between experimental results and DFT. That is, the nuclear spins see the electronic states at the Fermi level and relax via their excitations, while the temperature limits the available states through the Fermi function ($-k_BT$). As mentioned in the results section, the relaxation rate values in the CDW phase denote the average of the 3 individual resonance lines, and similarly the low temperature DOS in panel (a) are not resolved for the 3 V sites. It may thus be possible that the relaxation also shows differences among the 3 V sites similar to the shift values. This, however, could not reliably be resolved with experiment.

A different way to look at the relaxation data is presented in Fig. 5b where the $(T_1T)^{-1}(T)$ is separated into 3 parts, Fermi liquid behavior below and above the CDW phase transition separated by a change in DOS by about a factor of $\sqrt{2}$, and a high temperature dependence above 150 K that can very well be approximated by an activation-type of behavior. A single exponential fit yields a gap of the size of about 160 meV (cf. Arrhenius plot in panel c) which might be relatable to band structure features, such as van Hove singularities as suggested by others[41]. In the current case of ScV$_6$Sn$_6$, however, the band structure is too complex to unambiguously relate the spin-lattice relaxation to individual bands and their dispersion. Finally, our observation may also be related to the experimental results of a significant loss in carrier concentration during cooling, on the basis of which a pseudogap behavior has been proposed[42].

The EFG at V sites accessible through the V quadrupole interaction gives a direct evidence of a characteristic charge redistribution due to the CDW phase transition. As shown in the Fig. 3, the single high temperature vanadium EFG is replaced by 3 low temperature EFGs differing in size and shape, but retaining the original orientation in the crystal lattice. The 3 different EFGs mapped on the known low temperature crystal structure correspond to a CDW with wave vector $\mathbf{q} = \left(\frac{1}{3}, \frac{1}{3}, \frac{1}{3}\right)$ which agrees with the reported reconstruction of the unit cell with $\sqrt{3} \times \sqrt{3} \times 3$ periodicity. From the splitting frequencies we determine the phase to be $\varphi \approx \frac{\pi}{4}$ for 1D sinusoidal modulations along any crystal direction. The real-space charge distribution that leads to the different EFGs for V1 to V3 cannot be extracted from the experimental results. Plausibly, the EFGs could display the imbalance of Fermi level states in the five V 3$d$ orbitals related to the following discussion of the NMR shift. But they may also be affected by charges from bands below the Fermi energy and offsite contributions such as neighbored ions (e.g., the 3 Sn sites in Fig. 2)[43].

We further determined the NMR shift for two crystal orientations ($a^*$ and $c\|B_0$) and for a wide range of temperatures as shown in Fig. 4c, d. For the analysis, we may separate the shift into two main components: (1) a very large and highly anisotropic part with a slight temperature dependence, and (2), the characteristic splitting as a consequence of the CDW phase transition. Component (1) is most likely dominated by a very large orbital shift (from 0.6% to well above 0.8%) stemming from the magnetic moment associated with the orbital motion of the electrons (van Vleck paramagnetism) that was repeatedly found in metallic V compounds[38,44–48]. This shift term is typically independent of or weakly dependent on temperature, was reported to be insensitive to the CDW phase transition in 1T-VS$_2$, and may thus explain why the total shift including its temperature dependence seems not to be directly related to the relaxation as expected from a Korringa relation (from $K^2T_1T = const.$ we expect an isotropic shift of less than 0.1 % with a distinct temperature dependence)[38,48].

The second shift component (2) then contains all the changes due to the CDW phase. Obviously, the CDW phase emerges as an in good

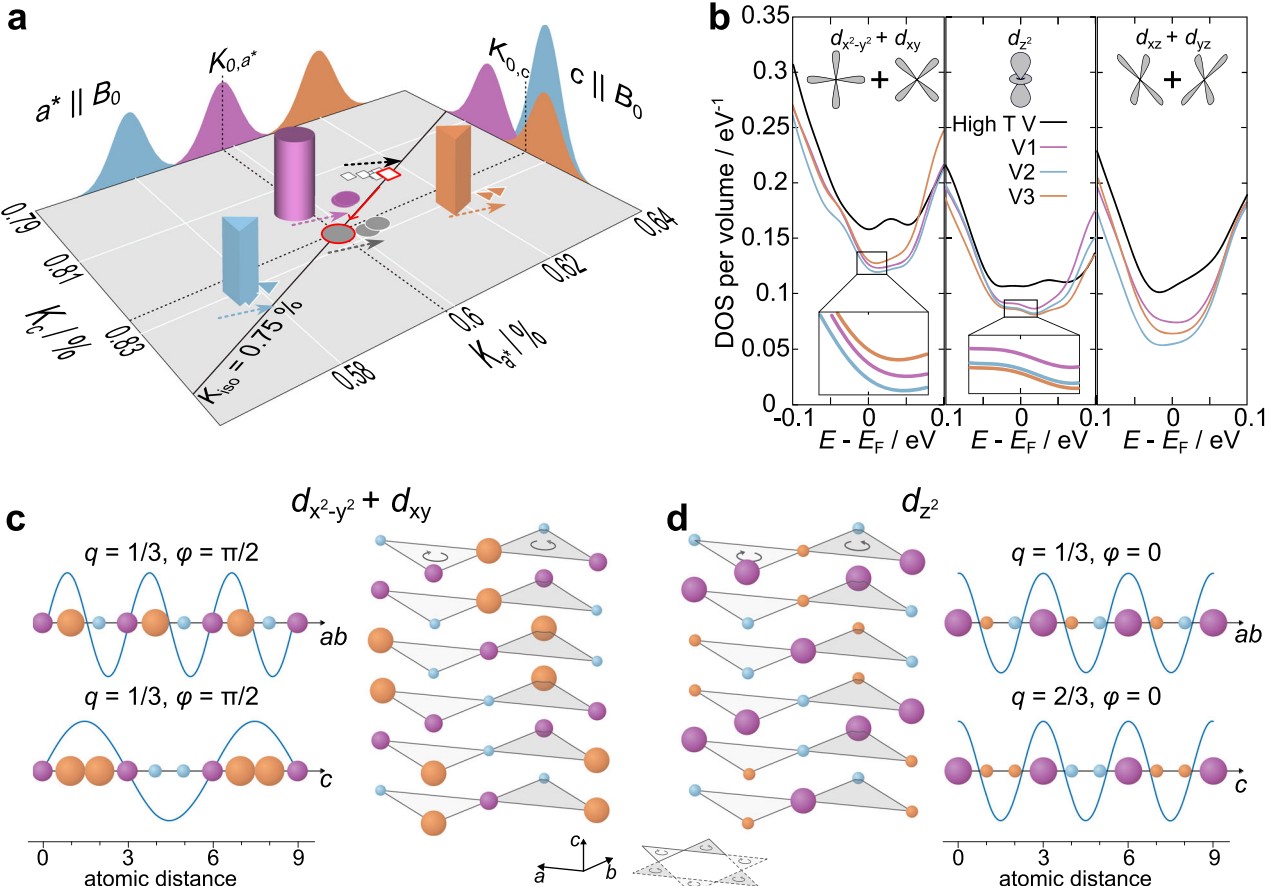

**Fig. 6 | Orbital selective modulations of the local density of states. a** The shift values from Fig. 4 (**a**) plotted as $K_c$ vs. $K_{a^*}$ with the temperature as an intrinsic parameter. The 3D objects denote the three central transitions (CTs) of V1, V2, and V3 at 80 K. When measured for $a^* \| B_0$, the three CTs are well separated, but for $c \| B_0$, V2 and V3 have the same frequency to form a twice as intense resonance, and, with V1 separated, the asymmetric doublet of Fig. 4 (**a**). The red edged white square corresponds to 96 K, the red edged gray circle also to 80 K. The dashed arrows indicate the direction of $K_i(T)$ for decreasing temperatures. The black diagonal marks a line of constant isotropic NMR shift of 0.75 %. **b** Orbital and site resolved density of states (DOS) for high (black) and low (colored) temperatures from

density functional theory (DFT). **c** The resulting sinusoidal modulation of the DOS for $d_{x^2-y^2} + d_{xy}$ orbitals (*left*) and the crystal $a$ and $b$ directions (equivalent) as well as for $c$, beside V hourglass building blocks to illustrate the structural stacking of the V kagome planes in the charge density wave (CDW) phase (*right*). Colors denote crystal sites, i.e., V1 to V3, while the size of the balls represent the local DOS. The curved arrows denote the handedness of the chiral stacking. **d** Similar to panel (**c**) for the $d_{z^2}$ orbitals and the corresponding modulation patterns of the local DOS. Obviously, the wavelength of the DOS modulation for $d_{z^2}$ orbitals is half the CDW unit cell in the crystal $c$-direction, yielding $\mathbf{q}_{z^2} = (\frac{1}{3}, \frac{1}{3}, \frac{2}{3})$.

approximation temperature independent splitting of the single high temperature resonance line into a symmetric triplet of resonances for $a^*$ and an asymmetric doublet for $c \| B_0$. The switching between the two types of splittings appears unusual, but especially the twofold splitting of three components in the $c \| B_0$ orientation is surprising, because it is not clear what causes such a symmetry breaking. To illustrate this puzzle, we show in Fig. 6a the shift data from Fig. 4 as a $K_c$ vs. $K_{a^*}$-plot. The two shift patterns observed for $c$ and $a^* \| B_0$ are sketched as projections on the two perpendicular axes (the 3d objects represent shift data at 80 K). They are fingerprints of commensurate 1D CDWs with $q = \frac{1}{3}$ that have a phase difference of $\Delta\varphi = \frac{\pi}{2}$ (cf. Fig. 1c, d). The only difference between the two shift splittings is the orientation of the external field and it is rather unlikely that the field directly induces a phase change in the corresponding CDW. This raises the question for what causes the observed phase difference?

The shift anisotropy as a function of temperature brings some light into this problem. Beside the individual temperature dependencies drawn on the plane in Fig. 6a (color coding follows Fig. 4c, d), the red edged data points, representing $K$ just above (white square - 96 K) and the averaged $K$ below (gray circle - 80 K) the CDW phase transition, are both located almost exactly on a line of constant isotropic shift of

0.75 %. Under the reasonable assumption of axially symmetric shift tensors for both temperature regimes, the average isotropic shift thus remains unaffected by the phase transition, while the average axial component substantially changes (Supplementary Fig. 7). Hence, the observed splittings in $a^*$ and $c \| B_0$ are mainly caused by a splitting in the local electronic spin susceptibility (Pauli paramagnetism) with an anisotropic hyperfine interaction. Contact interaction and core polarization effects that yield isotropic NMR shifts cannot be responsible for the peculiar shift splittings. Spin dipolar interactions, on the other hand side, are highly anisotropic and may offer a way to resolve the conundrum of the phase shift between the two different field orientations[38,49]. The shift measured for different field orientations could represent the shift contribution from orbitals with different geometry. In particular, the symmetric triplet in the plane may predominantly arise from different DOS in the planar orbitals, i.e., $d_{x^2-y^2}$ and $d_{xy}$, while the asymmetric doublet that appears with the field along the crystal $c$-axis displays the DOS of the orbital with a distinct component along that direction, i.e., $d_{z^2}$.

To gain independent insight, we examined the DOS of individual orbitals for the 3 different V sites using DFT. The results are presented in Fig. 6b resolved for the sum of the $d_{x^2-y^2} + d_{xy}$ orbitals (for

symmetry reason both in-plane orbitals are equivalent), for $d_{z^2}$, and for the $d_{xz} + d_{yz}$ orbitals (again, both orbitals are equivalent). The site-selective DOS for the in-plane orbitals at the Fermi level splits symmetrically into 3 different DOS (purple, orange, and light blue), while for the out-of-plane orbital $d_{z^2}$ the splitting is twofold, with V2 and V3 (orange and light blue) having almost exactly the same DOS. For $d_{xz} + d_{yz}$ the threefold splitting is again symmetric and visibly larger than for the other two components. Thus, DFT calculations almost perfectly reproduce the characteristic splittings observed with NMR, suggesting that the distinct orbital geometry is dominating the shift for special field orientations. Mapping the NMR/DFT data onto the low temperature crystal structure with V1, V2, and V3, we find the periodic, orbital specific modulations of the DOS as shown in panels (c) and (d) of Fig. 6. For $d_{x^2-y^2} + d_{xy}$ orbitals, the corresponding wave vector is $\mathbf{q}_{xy} = \left(\frac{1}{3}, \frac{1}{3}, \frac{1}{3}\right)$ with a phase $\varphi \approx \frac{\pi}{2}$. The modulation for $d_{xz} + d_{yz}$ orbitals (not shown) has the same wave vector but $\varphi \approx \frac{\pi}{6}$. Due to the twofold splitting in the $d_{z^2}$ orbitals, the corresponding modulation is characterized by $\mathbf{q}_{z^2} = \left(\frac{1}{3}, \frac{1}{3}, \frac{2}{3}\right)$ and $\varphi \approx 0$ (see also Supplementary Fig. 8). Thus, NMR and DFT imply orbital specific modulations of the DOS that differ in amplitude, phase, and, in case of the V $d_{z^2}$ modulation, even in periodicity of the out-of-plane component. In this scenario, the switching between the in-plane orbitals and the out-of-plane orbitals requires a highly selective hyperfine interaction. This should be subject of a theoretical investigation in future studies. The chirality of the crystal structure is transferred to the modulations of the DOS where corner-sharing triangular columns (light and dark gray) have opposite handedness. It should be noted that the in-plane patterns of the DOS modulations have the same unit cell as given by Arachchige et al. 2022[6]. We provide these patterns in Supplementary Fig. 8. The DOS pattern of the in-plane orbitals can be seen as three, by a factor of $\sqrt{3}$ enlarged and intersecting kagome lattices (the same holds for the $d_{xz} + d_{yz}$ orbitals as well as for the EFGs shown in Fig. 3f). Contrastingly, since the $d_{z^2}$ orbitals of V2 and V3 have approximately the same DOS, they form hexagons centered inside the enlarged kagome pattern of V1. This configuration resembles the one shown by Arachchige et al.[6].

A quantitative assessment of the scenario discussed above requires a detailed knowledge of the corresponding hyperfine interaction, which are notoriously difficult to calculate. That is, although typically isotropic, it cannot be excluded that contact interaction or core polarization contribute to the NMR shift and the observed changes, similarly the orbital shift component. From spin-lattice relaxation that most potentially reflects the electronic spin susceptibility only (not orbital), on the other hand side, we calculate 600 before (-100 K) and about 400 ppm after (-80 K) the phase transition based on the Korringa relation, $K^2 T_1 T = \left(\gamma_e / \gamma_n\right)^2 \left(\hbar / 4\pi k_B\right)$[50]. The -200 ppm implied changes from $T_1$ fit quite well the drop of the average shift (gray circles in Fig. 4c, d) during the phase transition, as well as the observed splittings. This, however, seems not exactly be reflected in the orbital resolved DOS, where the overall drop due to the phase transition is larger than the splitting, pointing at a more complex shift phenomenology. Similarly, the temperature dependence observed in the relaxation at higher temperatures (800 to 600 ppm between 300 and 100 K) implies a temperature dependent NMR shift which is only partly observed for $c \| B_0$ (about 100 ppm).

In summary, we investigated the CDW kagome metal ScV₆Sn₆ using single crystal ⁵¹V NMR and DFT. The CDW phase transition occurs between 96 and about 80 K and takes place as a gradual replacement of the high temperature phase by the CDW phase as expected from a first order transition. The phase transition is accompanied by a change in the electronic band structure and the corresponding changes in the total DOS by a factor of about $\sqrt{2}$ is very well reproduced with DFT to match the experimentally observed evolution of the local magnetic field fluctuations for decreasing temperatures. The CDW phase features an individual conversion of the three formerly equivalent V environments per V triangle in agreement with DFT and the reported

reconstruction of the unit cell with $\sqrt{3} \times \sqrt{3} \times 3$ periodicity. In the CDW phase, our findings further comprise an unusual orientation dependent change in the NMR shift splitting from a symmetric triplet of resonance lines for $a^* \| B_0$ to an asymmetric doublet for $c \| B_0$, while the latter reflects a symmetry that cannot be found in the crystal structure. When regarded as one-dimensional sinusoidal modulations of the local magnetic field, this observation implies a magnetic field induced phase shift of $\Delta \varphi = \frac{\pi}{2}$. To resolve this discrepancy, we argue on the basis of orbital selective modulations of the local DOS with in-plane wave vector $\mathbf{q} = \left(\frac{1}{3}, \frac{1}{3}\right)$ but different out-of-plane wave vector components and phases, driven by a field orientation selective hyperfine coupling. The latter calls for a quantitative theoretical treatment of the hyperfine interaction and possible implications for the compound's transport properties. Our work demonstrates that the combination of single crystal NMR and DFT calculations leads to crucial local information about static and dynamic electronic properties in the charge density wave system ScV₆Sn₆.

## Methods

### Crystal synthesis
High-quality single crystals of ScV₆Sn₆ were grown by the flux method[6]. High-purity Sc, V, and Sn elements were loaded in an alumina crucible in a molar ratio of 1 : 10 : 60 and then sealed in a quartz tube under vacuum. The tube was then slowly heated to 1373 K, maintained for 10 h, and cooled down to 973 K over 400 h. Hexagonal shape crystals with silvery surfaces and a typical size of $2 \times 2 \times 1$ mm³ were obtained after centrifugation. The crystal structure was refined from powder x-ray diffraction and the chemical components were examined by using energy-dispersive X-ray spectroscopy. The crystal orientations were determined by using the Laue backscattering diffractometer.

### NMR experiments
Measurements were carried out on a JANIS sweepable 9 T magnet and a TECMAG APOLLO NMR console. Most experiments were carried out on the hexagonal, plate-like Sample S2 with dimensions $1.7 \times 1.1 \times 0.2$ mm³ (cf. Yi et al. 2024[16]). A few measurements were done on Sample S1 with $1.5 \times 0.45 \times 0.15$ mm³. In both cases, the rf-coil was wound directly around the sample and placed on the single axis goniometer (accuracy ~-1°) of a home-built probe. Most experiments were carried out at 8.73 T typically using broad-band free induction decay (FID) measurements with pulse lengths of 0.5 μs in combination with low $Q$-factors (in the order of 10), as well as selective FID or spin-echo $\left(\frac{\pi}{2} - \tau - \pi\right)$ experiments with $\frac{\pi}{2}$-pulse lengths of 5 or 10 μs for individual resonance lines ($Q$ between 16 and 32, cf. Supplementary information). Selective saturation recovery pulse sequence $\left(\frac{\pi}{2} - \Delta - \text{FID/spin echo}\right)$ was employed to measure the spin-lattice relaxation time $T_1$ of individual transitions. The shifts were obtained by referencing the ⁵¹V resonance frequencies versus VOCl₃ using the second reference method[51] and the omnipresent ⁶³Cu resonance line of the rf-coil.

### Numerical calculations
The simulated results were obtained by using ab initio calculations in the framework of density-functional theory (DFT), as implemented in the program VASP[39]. In this code, augmented plane waves are used as a basis set together with pseudopotentials. To describe the exchange-correlation potential, the generalized-gradient approximation (GGA)[52] was used. The calculations are based on the structural models provided by Arachchige et al.[6].

The EFGs and self-consistent calculations were carried out on a $18 \times 18 \times 9$ ($6 \times 6 \times 6$) k mesh for the high (low) temperature phase, respectively. Convergence for total energy and EFGs was carefully checked. For the DOS calculations, a k mesh of $33 \times 33 \times 33$ ($19 \times 19 \times 19$) was used for the high (low) temperature phase, respectively.

## Data availability

All data supporting the findings of this study are available within the article and the Supplementary Information file. Raw data generated during the current study are available from the corresponding authors upon request.

## Code availability

The codes that support the findings of this study are available from the corresponding authors upon request.

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

## Acknowledgements

The authors thank O. Stockert, S. Wirth, J. Sichelschmidt, X. Feng, W. Schnelle, and J. Haase (Leipzig) for helpful discussions. We acknowledge the financial support by the Deutsche Forschungsgemeinschaft (DFG) under SFB1143 (Project No. 247310070), the Würzburg-Dresden Cluster of Excellence on Complexity and Topology in Quantum Matter-ct.qmat (EXC 2147, Project No. 390858490). M.G.V. acknowledges support to the Spanish Ministerio de Ciencia e Innovacion (grant PID2022-142008NB-I00), partial support from European Research Council (ERC) grant agreement no. 101020833, the European Union NextGenerationEU/PRTR-C17.I1, by the IKUR Strategy under the collaboration agreement between Ikerbasque Foundation and DIPC on behalf of the Department of Education of the Basque Government and the Ministry for Digital Transformation and of Civil Service of the Spanish Government through the QUANTUM ENIA project call - Quantum Spain project, and by the European Union through the Recovery, Transformation and Resilience Plan - NextGenerationEU within the framework of the Digital Spain 2026 Agenda. M.G.V. and C.F. acknowledge funding from the Deutsche Forschungsgemeinschaft (DFG, German Research Foundation) for the project FOR 5249 (QUAST).

## Author contributions

R.G. performed NMR experiments, analyzed data, and wrote the manuscript. M.B. assisted in the NMR experiments and worked with R.G. on the interpretation of the NMR data. J.N. performed theoretical calculations. J.N. and M.G.V. assisted in data analysis. C.Y. and C.S. grew and characterized single crystals. C.F. supervised the project. All authors commented on the manuscript.

## Funding

## Competing interests

The authors declare no competing interests.
