## [Peer Review File · Nature Communications]

REVIEWER COMMENTS

Reviewer #1 (Remarks to the Author):

Since identifying superconductivity in CsV₃Sb₅, the kagome system has garnered significant attention within the scientific community. Another compound, ScV₆Sn₆, possessing kagome V-planes, exhibits a charge density wave (CDW) order similar to CsV₃Sb₅. This paper offers a comprehensive analysis of the CDW structure in ScV₆Sn₆ through a 51V nuclear magnetic resonance (NMR) investigation and gives reliable results. I recommend publishing this manuscript in NC. Below are my comments and suggestions.

1) In the introduction, the authors say that CsV₃Sb₅ is an unconventional superconductor without any reference. But as far as I know, it is a conventional superconductor with unconventional CDW order. If they insist on this view, references should be provided.

2) Ref [29] gives a wrong CDW structure (star-of-David) and their data are suspicious. Actually, it is Tri-hexagonal structure proved by other experiments and theory analysis (Chinese Phys B 31, 017105(2022), Phys. Rev. B 107, 184106(2023), Physical Review Research 5, L012017(2023).). I suggest deleting this reference to avoid misleading or adding others as comparison references.

3) If using the definition of Eq.(1), q should be $(2\pi/3a, 2\pi/3a, 4\pi/3a)$. Otherwise, Eq.(1) should be $\rho = \rho_0 + \Delta \rho \cos(\frac{2\pi}{a} q x + \phi)$.

4) In Eq.(2) and Eq.(3), it should be $\cos(2\alpha)$, as the Eq.(1) in supplement. (G. H. Stauss, J. Chem. Phys. 40, 1988(1964).)

5) Why is the first satellite peak at around 0.5 MHz stronger than the central peak at 80 K in Fig. 3(a)? The central peak has the same shape and intensity as the second satellite peak at around 0.8 MHz. Is it because of experimental error? Moreover, the first satellite peak at 100 K is too small while the baseline noise is smaller than others, in Fig S1.

6) There is interference with the copper signal from the coil, in Fig S5, which masks the lower part of the V quadrupole split in Fig 3 (e). This can be overcome by using a silver coil.

7) The unit cell contains 2 V atoms along a-axis as shown in Fig. 1. Why there is only 1 V atom in a unit cell in Fig.6 (c) and (d)? The modulation pictures are not compatible with Ref.[5].

8) Can this experiment distinguish between $(1/3, 1/3, 1/3)$ and $(1/3, 1/3, 2/3)$ wavevectors?

Reviewer #2 (Remarks to the Author):

This is a comprehensive study of the kagome metal ScV_6Sn_6 via NMR. The material under examination is interesting as it is a vanadium-based kagome metal with CDW order, similar to the AV_3Sb_5 ($A=\text{K}, \text{Rb}, \text{Cs}$) compounds. Although the CDW vector in ScV_6Sn_6 is completely different from those in AV_3Sb_5 compounds, time-reversal symmetry (TRS) is suggested to be broken in both systems. The present work finds (1) an absence of local moments that could be associated with TRS-breaking, (2) an absence of signatures for short-range $(1/3, 1/3, 1/2)$ CDW, (3) sixfold in-plane rotational symmetry in the ab -plane, and (4) an orbital-selective CDW modulation, based on the orientation dependence of the NMR splitting. Points (1)-(3) are important for topics such as TRS-breaking and nematicity in the kagome metals, but it is unclear whether the present results provide convincing information that further the understanding of these issues. Point (4) is highly tantalizing, however the evidence for such a picture seems rather indirect.

Overall, this work examines a material and associated physical properties of general interest, but does not appear to present significant advances in a convincing manner, and is not suitable for publication in the present form. The manuscript could be reconsidered, once the authors address the following issues:

TRS-breaking is often subtle, and conflicting observations are often attributed to varying sensitivity of different techniques. Can the authors estimate the local moments needed to see appreciable signals in NMR, and set an upper limit for the moment due to TRS-breaking? The stated 100 ppm in terms of the linewidth is not very helpful for most readers.

Sixfold rotational symmetry is found to be retained in the CDW state. Can this work differentiate between intrinsic sixfold symmetry and apparent sixfold symmetry due to twinning? Statements on this front will be helpful as nematicity is an important theme in the study of AV_3Sb_5 .

ScV_6Sn_6 is peculiar from the perspective of CDW formation, with the $(1/3, 1/3, 1/3)$ CDW order occurring via a first-order transition, despite phonon softening (flat-mode-like, and is led by the vector $(1/3, 1/3, 1/2)$) which usually precedes a second-order transition, occurring just above TCDW. This work does not find evidence for the $(1/3, 1/3, 1/2)$ short-range CDW above TCDW, but as these are slow fluctuations, is NMR sensitive to such slowly fluctuating dynamic CDW?

What kind of real space distribution of charge density would the orbital-selective CDW modulation lead to? Is this consistent with the known CDW structure from single crystal X-ray diffraction? The extraction of the phase information refers to 1D CDWs, how does this translate to the realistic material, which is 3D? The CDW modulation of the d_{z^2} orbital is suggested to be described by

(1/3,1/3,2/3), how is this different from (1/3,1/3,1/3), and would this lead to additional periodicity not seen in diffraction?

An activation gap of 160meV is suggested, what kind of gap is this, how would it show up in other physical properties? Could this be related to the pseudogap behavior seen in npj Quantum Materials 8, 65 (2023)?

Minor issues:

The reduction of the DOS by a factor of $\sqrt{2}$ is mentioned several times. This seems to suggest a very precise knowledge on the quantitative value (as compared to say ~ 1.4), do the data or calculations back up such a precise value?

In Fig. 2(b), more atoms can be labelled by V1, V2, V3, to illustrate the periodicity of the CDW.

In Fig. 3(f), it is helpful to show a unit cell for the CDW phase, to illustrate how the periodicity repeats.

Reviewer #3 (Remarks to the Author):

The manuscript Orbital selective commensurate modulations of the local density of states in ScV6Sn6 probed by nuclear spins reports 51V NMR spectroscopy and relaxation measurements carried out over a wide temperature range, and covering the first order phase transition to a charge-ordered phase at about 90 K.

The spectroscopy is consistent with prior reports of a commensurate ordering wavevector, $Q=(1/3,1/3,1/3)$, which sets in over a varying volume fraction as the crystal under study is cooled through the range 96 K to 80 K. To account for measured spectrum of the low temperature phase, the authors need to include in their analysis a choice of the CDW phase relative to the lattice periodicity, and the choice is different depending on the field orientation, $B//a^*$ or $B//c$. They attribute this apparent discrepancy by noting that the relative contributions to the shifts by the various 3d orbitals will naturally depend on the applied field direction.

The relaxation data gives good supporting evidence for a discontinuous drop in DOS in the charge-ordered phase that is consistent with expectations from DFT calculations.

The presentation is thorough as well as somewhat technical. Perhaps this is unavoidable under the circumstances. But it also leads me to consider that Nature Communications may not be the most appropriate venue. Countering that perspective is the widespread attention on Kagome metals, which host topologically non-trivial electronic structures and the possible consequences to the broken-symmetry phases observed in systems such as AV3Sb5 and ScV6Sn6. The current results will certainly provide information relevant to those considerations, and thus warrant publication in Nature Communications.

Some further points that should be considered prior to publication:

1. The explanation provided to explain the spectrum $T < T^*$ is plausible at a qualitative level. However, the manuscript would be more complete if quantitative information of the shift tensor for normal and charge-ordered regimes are included, and compared to the fractional contribution of the different orbitals indicated by the DFT. It could be step toward estimating the orbital contributions.
2. The variation of $1/T1T$ over the temperature range 200-300 K, is not correspondingly showing up in the shift data $B//c$ ($B//a^*$ is not reported in Fig. 4d). The authors note the result but do not comment further on its origin.
3. The authors insist that their measurements provide no support for reports of Time Reversal Symmetry Breaking (e.g., μ SR, Ref. 24). However, there is no substantial quantitative analysis of the present sensitivity relative to expectations derived from those earlier findings. The authors should provide that information.

Detailed response to the reviewers

In the following a detailed point-by-point response to the review reports is provided. Any corresponding changes to the manuscript are summarized in the **List of changes** (LOC) thereafter, as well as changes due to formatting requirements.

Reviewer #1 (Remarks to the Author):

R1: Since identifying superconductivity in CsV_3Sb_5 , the kagome system has garnered significant attention within the scientific community. Another compound, ScV_6Sn_6 , possessing kagome V-planes, exhibits a charge density wave (CDW) order similar to CsV_3Sb_5 . This paper offers a comprehensive analysis of the CDW structure in ScV_6Sn_6 through a 51V nuclear magnetic resonance (NMR) investigation and gives reliable results. I recommend publishing this manuscript in NC. Below are my comments and suggestions.

A: Thanks you very much.

R1: 1) In the introduction, the authors say that CsV_3Sb_5 is an unconventional superconductor without any reference. But as far as I know, it is a conventional superconductor with unconventional CDW order. If they insist on this view, references should be provided.

A: The referee is right to point this out. Actually, a consensus of the nature of superconductivity seems not yet to be reached (a case pro unconventional SC for example in the paper cited by the referee in the next point, Physical Review Research 5, L012017(2023)). We have changed that this point in the manuscript accordingly. Point 6 in LOC.

R1: 2) Ref [29] gives a wrong CDW structure (star-of-David) and their data are suspicious. Actually, it is Tri-hexagonal structure proved by other experiments and theory analysis (Chinese Phys B 31, 017105(2022), Phys. Rev. B 107, 184106(2023), Physical Review Research 5, L012017(2023)). I suggest deleting this reference to avoid misleading or adding others as comparison references.

A: We reassessed the mentioned paper and decided to add other works and clarification in the introduction to avoid potential misleading. Point 7 in LOC.

R1: 3) If using the definition of Eq.(1), q should be $(2\pi/3a, 2\pi/3a, 4\pi/3a)$. Otherwise, Eq.(1) should be $\rho = \rho_0 + \delta\rho \cos(\frac{2\pi}{a}qx + \phi)$.

A: We thank the referee for hinting at this inaccuracy. For convenience we assumed $2\pi/a = 1$, however, without further explanation (and with a wrong definition of λ). We updated the manuscript in this respect. Point 8 in LOC.

R1: 4) In Eq.(2) and Eq.(3), it should be $\cos(2\alpha)$, as the Eq.(1) in supplement. (G. H. Stauss, J. Chem. Phys. 40, 1988(1964)).*

A: Thank you very much! We corrected the equations. Point 9 in LOC.

R1: 5) Why is the first satellite peak at around 0.5 MHz stronger than the central peak at 80 K in Fig. 3(a)? The central peak has the same shape and intensity as the second satellite peak at around 0.8 MHz. Is it because of experimental error? Moreover, the first satellite peak at 100 K is too small while the baseline noise is smaller than others, in Fig S1.

A: The "unusual" appearance of the low temperature spectrum is the result of the combination of individual changes in the shift and quadrupole splitting due to the CDW. For the first high frequency satellite, the shifts of the 3 Vanadium nuclei (V1, V2, V3 - which gives rise to the asymmetric central line) are compensated by smaller/larger quadrupolar splittings, such that the 3 satellites line up, yielding the rather narrow resonance in comparison with its neighbors.

Fig. 3a and supplementary figure 1 show combined spectra from individually (selective, transition-wise) measured resonances for each temperature. The relative signal intensity for such a quadrupole pattern is well defined, however, requires well adjusted experimental parameters, which may not have perfectly been met. The difference in the noise levels for individual resonances in Supplementary Fig. 1 (formerly S1) reflects different

signal averaging (number of averaged experiments/scans and thus signal-to-noise ratio) of the various NMR signals. The spectra as presented in Fig. 3(a) and Supplementary Fig. 1 are corrected (not arbitrarily normalized) for signal averaging, temperature, and the rf-circuit's quality factor, and are therefore directly comparable. Limitations in the adjustments of the experimental parameters can lead to losses in signal intensity. However, we can account for a consistency of the total signal intensity, as we have shown in the Supplementary Fig. 9 (formerly S8). We clarified these details in the Supplementary discussion. Point 11 in LOC.

R1: 6) There is interference with the copper signal from the coil, in Fig S5, which masks the lower part of the V quadrupole split in Fig 3 (e). This can be overcome by using a silver coil.

A: The referee is right, using an rf-coil made from silver wire would remove the ^{63}Cu signal. However, beyond the ^{63}Cu signal, in that central range of about 500 kHz, there overlap in total 21 resonances (3×7) of about 10 kHz linewidth (belonging to nuclei with the field along V_{xx}) plus the 6 central lines of the other patterns. We decided that an unambiguous and convincing resonance selection and assignment is not possible, and therefore, the corresponding splittings could not be extracted.

R1: 7) The unit cell contains 2 V atoms along a-axis as shown in Fig. 1. Why there is only 1 V atom in a unit cell in Fig.6 (c) and (d)? The modulation pictures are not compatible with Ref.[5].

A: Thank you very much for pointing this out. After double-checking the mentioned figure, we find the way the unit cell is depicted is at least misleading. The initial purpose was to emphasize the changes and the different wavelengths of the modulation, especially with respect to the modulation along c . We removed these details, however. Concerning the compatibility with Ref.[5] (now [6]), the patterns of the orbital and site selective DOS have the same unit cell as reported by [5]. We understand that the manuscript lacks a proper comparison and thank the referee for pointing this out. We added a discussion of our DOS patterns with respect to reference to [5] and provide a new figure (Supplementary Fig. 8) with the in-plane configurations. The out-of-plane configuration, however, is difficult to extract from [5]. See point 12 in the LOC.

R1: 8) Can this experiment distinguish between $(1/3, 1/3, 1/3)$ and $(1/3, 1/3, 2/3)$ wavevectors?

A: The identification of the wavevector $(1/3, 1/3, 2/3)$ relies on the known crystal structure and the consistent NMR evidence of the three different V sites, V1, V2, V3 (quantitative agreement with DFT). For the external field along the crystal c -axis, V2 and V3 have the same shift. Due to the arrangement of the V chains along the crystal c -axis, a modulation appears with the wavevector $(1/3, 1/3, 2/3)$. The NMR experiment "only" gives us a histogram of the local nuclear environments (typically chemically non-equivalent sites), i.e., how many different surroundings are there, and how often do they occur, plus some symmetries (three-fold splitting + sixfold rotation symmetry in the plane). What the experiment clearly demonstrates is that the shift pattern for $c \parallel B_0$ (asymmetric doublet) breaks the threefold symmetry of the crystal structure.

Reviewer #2 (Remarks to the Author):

R2: This is a comprehensive study of the kagome metal ScV_6Sn_6 via NMR. The material under examination is interesting as it is a vanadium-based kagome metal with CDW order, similar to the AV_3Sb_5 ($A=K,Rb,Cs$) compounds. Although the CDW vector in ScV_6Sn_6 is completely different from those in AV_3Sb_5 compounds, time-reversal symmetry (TRS) is suggested to be broken in both systems. The present work finds (1) an absence of local moments that could be associated with TRS-breaking, (2) an absence of signatures for short-range $(1/3,1/3,1/2)$ CDW, (3) sixfold in-plane rotational symmetry in the ab-plane, and (4) an orbital-selective CDW modulation, based on the orientation dependence of the NMR splitting. Points (1)-(3) are important for topics such as TRS-breaking and nematicity in the kagome metals, but it is unclear whether the present results provide convincing information that further the understanding of these issues. Point (4) is highly tantalizing, however the evidence for such a picture seems rather indirect.

A: Thank you very much for the detailed assessment of our manuscript. Concerning the evidence of our proposed orbital selective DOS being direct or indirect, please consider the following: the body of our NMR data in the present system gives a consistent picture of the local electronics, as, (1) the 3 chemically non-equivalent V sites are reflected in the 3 different quadrupole splittings (EFGs) and their relative number of 1:1:1, which is in very good agreement with DFT; (2) the local magnetic field (NMR shift) divides the environments also in 3 different components (with a directly accessible pattern in $c \parallel B_0$ that clearly breaks the threefold symmetry of the crystal structure), each of them assignable to one of the quadrupole splittings; (3) the changes in the spin-lattice relaxation reflect those of the total DOS (in quantitative agreement with DFT), while the drop during the phase transition (by about 0.05 (sK)^{-1}) fits very well the changes/splitting of the NMR shift calculated from the Korringa relation ($\Delta(T_1T) = 0.05 \text{ (sK)}^{-1} \Rightarrow \Delta K \approx 200 \text{ ppm}$). Thus, relationship to the quadrupole splitting proves the local magnetic field to be modulated on a atomic level, i.e., the local field alternates from neighbor to neighbor, while the quantitative agreement with the relaxation connects the modulation to the local DOS, as reflected in the orbital and site resolved DOS from DFT calculations.

Consequently, the observation of different splitting symmetries for different field orientations as discussed in our manuscript imply either, (a) the external field *changes directly* the relative DOS for different crystal sites, i.e., a magnetic field induced phase shift in the modulation, or, (b) the field *selects* between different orbitals and their respective modulations via hyperfine interaction and their anisotropy, implying orbital selective DOS modulations. Both explanations are really interesting. In the light of the overall consistent data and the straightforward analysis of our experimental and numerical results, orbital and site selective DOS provide a coherent explanation.

R2: Overall, this work examines a material and associated physical properties of general interest, but does not appear to present significant advances in a convincing manner, and is not suitable for publication in the present form. The manuscript could be reconsidered, once the authors address the following issues:

R2: TRS-breaking is often subtle, and conflicting observations are often attributed to varying sensitivity of different techniques. Can the authors estimate the local moments needed to see appreciable signals in NMR, and set an upper limit for the moment due to TRS-breaking? The stated 100 ppm in terms of the linewidth is not very helpful for most readers. Sixfold rotational symmetry is found to be retained in the CDW state. Can this work differentiate between intrinsic sixfold symmetry and apparent sixfold symmetry due to twinning? Statements on this front will be helpful as nematicity is an important theme in the study of AV_3Sb_5 .

A: Thanks for this comment. It is rather difficult to estimate an upper limit without the knowledge of the nature of the magnetism and thus potential interactions with nuclei (hyperfine coupling). However, when assuming a localized moment at the center of the V hexagonal voids, the corresponding dipolar field can be used. 100 ppm corresponds to about 1 mT, yielding a localized magnetic moment of well below $0.01 \mu_B$ to remain undetectable with our current method. Most potentially, the reported moments detected by muon-spin rotation spectroscopy may be below the resolution of NMR. We improved this part of our discussion, Point 13 in LOC.

A: Concerning the ability to differentiate *between intrinsic sixfold symmetry and apparent sixfold symmetry due to twinning*. From the mere NMR point of view we cannot distinguish intrinsic 6-fold symmetry from an apparent 6-fold symmetry due to twinning, because NMR gives a histogram of the local nuclear surroundings, but not necessarily information about whether these surroundings are direct neighbors or distant on a macroscopic level. We admit that it was not made clear enough in the main manuscript, that the six-fold rotational symmetry as reflected by the NMR is a consequence of the three-fold symmetry of the lattice (single Kagome layer) and the inability for NMR to differentiate mirror images with respect to the a^*c -plane, i.e., in the current case, NMR cannot distinguish between a 60° and a 120° rotation, since from the structural point of view, the latter is the mirror image if the former. We clarified this point in the manuscript, cf. 14 in LOC.

R2: ScV_6Sn_6 is peculiar from the perspective of CDW formation, with the $(1/3,1/3,1/3)$ CDW order occurring via a first-order transition, despite phonon softening (flat-mode-like, and is led by the vector $(1/3,1/3,1/2)$) which usually precedes a second-order transition, occurring just above TCDW. This work does not find evidence for the $(1/3,1/3,1/2)$ short-range CDW above TCDW, but as these are slow fluctuations, is NMR sensitive to such slowly fluctuating dynamic CDW?

A: The proposed CDW is associated with lattice vibrations along the crystal c -axis, with an emphasis on the trigonal Sn atoms that hover above/below the hexagonal V voids [11,13]. Slow fluctuations or the proposed vibrational modes of these Sn atoms can, theoretically, affect the static local charge symmetry (EFG) at V, yielding an additional component for the EFG that vanishes with the disappearance of the CDW and/or a characteristic broadening reflecting a distribution of the EFG component. Both of these signatures are not observed. The proposed CDW may also influence the dynamics of the V NMR. If the lattice vibrations match the quadrupole splitting frequency (500 kHz) or if the fluctuations of the local field associated with the CDW are close to the resonance frequency (100 MHz), the CDW could provide an additional relaxation channel, yielding a characteristic temperature dependence of the spin-lattice relaxation. Our results do offer any peculiarities that point in such a direction (spin-lattice relaxation is driven by free carriers, while T_1 during the phase transition is in very good agreement with the relative volume fractions for high and low T phase as obtained from the NMR signal intensities). We improved the discussion in this context. Point 15 in LOC.

R2: What kind of real space distribution of charge density would the orbital-selective CDW modulation lead to? Is this consistent with the known CDW structure from single crystal X-ray diffraction? The extraction of the phase information refers to 1D CDWs, how does this translate to the realistic material, which is 3D? The CDW modulation of the dz^2 orbital is suggested to be described by $(1/3,1/3,2/3)$, how is this different from $(1/3,1/3,1/3)$, and would this lead to additional periodicity not seen in diffraction?

A: We depicted two real-space patterns of the two different orbital specific DOS modulations in Fig.6 c and d. In order to provide an improved view of these patterns in the Kagome plane, we added another figure in the supplementary information (Point 12 in the LOC and Supplementary Fig. 8). It should be emphasized that we map the local field as obtained from the NMR histogram to the known crystal structure (via the quadrupole splittings and DFT). In particular, we follow reference [5] (now [6] and its supplementary information) in the original manuscript, that obtain their information from powder and single crystal X-ray diffraction. Our results are thus in agreement with the reported structure, i.e., the DOS patterns have the same in-plane unit cell as given by [5].

Using 1D chains has several purposes. It is directly relatable to the original model assumed by Peierls. 1D chains are further instructive to discuss modulations of the electronic density and their effect on NMR (Fig. 1). The 3D structure might then be assumed as an arrangement of parallel 1D V chains along the crystal c -axis in accordance with the crystal structure, i.e., the three neighboring chains forming a triangular V column are shifted with respect to each other by $1/3$ of the unit cell (2 V Kagome layers). This perspective transfers the chirality of the crystal structure along the c -axis to the charge order. A similar approach is for example given by Wezel and Littlewood, *Physics* 3, 87 (2010).

The wavelength of the associated modulation in c direction in real-space is half the unit cell (the latter being related to $\mathbf{q} = (1/3, 1/3, 1/3)$). Whether the difference in periodicity shows up in diffraction is questionable since very small differences in the electronic density would need to be distinguished from the background.

R2: An activation gap of 160meV is suggested, what kind of gap is this, how would it show up in other physical properties? Could this be related to the pseudogap behavior seen in npj Quantum Materials 8, 65 (2023)?

A: The spin-lattice relaxation rate is increasing non-linearly for temperatures above about 150 K. Since the relaxation is driven by free carriers, such a behavior implies an increase of states that were unavailable at lower temperatures. Since the band structure does not change (in the high temperature regime), these additional states must be gapped, and their availability then described by an activation law. Conceptually, this is similar to the observation of the superconducting gap from the spin-lattice relaxation measurements of superconductors below T_c , but it differs from the pseudogap observed with NMR in cuprates. A similar approach is used in ref. [44] of the original manuscript, the authors of which relate a sub-linear resistivity to contributions from a gapped (~ 80 meV) van-Hove singularity. In our case, we cannot assign the gap of 160 meV to a particular feature in the band structure and therefore associate the increase of the relaxation rate to combined effects of several band structure components. Theoretically, the gap should also show up in the electronic contribution of heat capacity, but due to the high temperatures, phonons potentially obscure a measurement. Thermopower is another method that should provide evidence of such a gap. Finally, it cannot be excluded that our observation is related to that made by the group mentioned above (npj Quantum Materials 8, 65 (2023)) that report a

significant loss of carrier concentration during cooling way above the CDW phase transition. Qualitatively, this is what we observe. We added this point in the manuscript. Point 17 in LOC.

R2: Minor issues:

R2: The reduction of the DOS by a factor of $\sqrt{2}$ is mentioned several times. This seems to suggest a very precise knowledge on the quantitative value (as compared to say 1.4), do the data or calculations back up such a precise value?

A: We used this value to emphasize the square root relationship between the spin-lattice relaxation and the DOS from theory. The experimental data, i.e., $(T_1T)^{-1}$, visibly drops in very good approximation by a factor of 2 during the phase transition, the corresponding changes in DOS would then be $\sqrt{2}$. Indeed, more precise values from experiment (1.40(1) for $T \rightarrow 0$) and DFT (1.44 at the Fermi level) are not exactly the same, but certainly close enough to constitute a connection between DOS and relaxation. We understand that this was not sufficiently made clear in the manuscript, and changed this point accordingly. Point 18 in LOC.

R2: In Fig. 2(b), more atoms can be labeled by V1, V2, V3, to illustrate the periodicity of the CDW.

A: Thanks, this is a very good recommendation. Point 19 in LOC.

R2: In Fig. 3(f), it is helpful to show a unit cell for the CDW phase, to illustrate how the periodicity repeats.

A: Thanks again. We followed the referee's suggestion. Point 20 in LOC.

Reviewer #3 (Remarks to the Author):

R3: The manuscript Orbital selective commensurate modulations of the local density of states in ScV₆Sn₆ probed by nuclear spins reports 51V NMR spectroscopy and relaxation measurements carried out over a wide temperature range, and covering the first order phase transition to a charge-ordered phase at about 90 K. The spectroscopy is consistent with prior reports of a commensurate ordering wavevector, $Q=(1/3,1/3,1/3)$, which sets in over a varying volume fraction as the crystal under study is cooled through the range 96 K to 80 K. To account for measured spectrum of the low temperature phase, the authors need to include in their analysis a choice of the CDW phase relative to the lattice periodicity, and the choice is different depending on the field orientation, $B//a^$ or $B//c$. They attribute this apparent discrepancy by noting that the relative contributions to the shifts by the various 3d orbitals will naturally depend on the applied field direction. The relaxation data gives good supporting evidence for a discontinuous drop in DOS in the charge-ordered phase that is consistent with expectations from DFT calculations. The presentation is thorough as well as somewhat technical. Perhaps this is unavoidable under the circumstances. But it also leads me to consider that Nature Communications may not be the most appropriate venue. Countering that perspective is the widespread attention on Kagome metals, which host topologically non-trivial electronic structures and the possible consequences to the broken-symmetry phases observed in systems such as AV₃Sb₅ and ScV₆Sn₆. The current results will certainly provide information relevant to those considerations, and thus warrant publication in Nature Communications.*

A: Thank you very much for the thorough review of our manuscript.

R3: Some further points that should be considered prior to publication:

R3: 1. The explanation provided to explain the spectrum $T < T^$ is plausible at a qualitative level. However, the manuscript would be more complete if quantitative information of the shift tensor for normal and charge-ordered regimes are included, and compared to the fractional contribution of the different orbitals indicated by the DFT. It could be step toward estimating the orbital contributions.*

A: Thanks for pointing this out. We altered the discussion to address a more quantitative shift assessment. Point 21 in LOC.

R3: 2. The variation of $1/T1T$ over the temperature range 200-300 K, is not correspondingly showing up in the shift data $B//c$ ($B//a^$ is not reported in Fig. 4d). The authors note the result but do not comment further on its origin.*

A: Thanks for this comment. We included the temperature dependence of the relaxation and the corresponding shift in the discussion, Point 21 in LOC.

R3: 3. The authors insist that their measurements provide no support for reports of Time Reversal Symmetry Breaking (e.g., μ SR, Ref. 24). However, there is no substantial quantitative analysis of the present sensitivity relative to expectations derived from those earlier findings. The authors should provide that information.

A: This point touches the comment of reviewer 2 on the evidence of time-reversal symmetry breaking. It is true that a quantitative analysis of the limitations of NMR to detect such an unusual magnetism is difficult and the given threshold 100 ppm perhaps insufficient. We therefore added an example of an upper limit of a localized moment in the Kagome plane to remain undetectable with our setup. Point 13 in LOC.

List of changes

1. We updated reference [1].
2. We removed "(as for ^{51}V NMR in CsV_3Sb_5)" in the "CDW and NMR" section of the results.
3. We added "(transition temperature from current study)" at the beginning of the "Structure" section of the results.
4. We corrected " $\sqrt{2}$ " to "2" in the discussion of the relaxation.
5. We added "The calculations are based on the structural models provided by Arachchige et al. 2022 [6]" and the end of the Charge symmetry section as well as in the methods.
6. We exchanged "an unconventional superconductor" by "superconducting" and added a citation in the first paragraph of the introduction.
7. We replaced "Similarly, NMR of ... and elevated pressure." by "Similarly, numerous NMR studies of CsV_3Sb_5 explored the microscopic structure of the Kagome lattice in the CDW phase, with a strong evidence for a tri-hexagonal configuration of the V atoms" and added 3 references in the third paragraph of the introduction.
8. We removed "with wavelength λ " before, and added "in units of $\frac{2\pi}{a}$ " after equation (1).
9. We changed " $\cos \alpha$ " to " $\cos 2\alpha$ " in equation (2) and (3).
10. We added "the" in the first paragraph on page 8.
11. We added "The individual noise levels reflect different signal averaging. The spectra were corrected for the temperature, signal averaging, and for the rf-circuit's quality factor. The two lowermost spectra (50 and 20 K) are further rescaled by $\times 2$ for clarity." in the caption of Supplementary Fig. 1.
12. We removed the boxes in Fig. 6 c and d, as well as "The black solid and dashed boxes denote the unit cell for the high temperature and the CDW structure, respectively." in the caption. We further added at the end of the discussion on page 16 "It should be noted that the in-plane patterns of the DOS modulations have the same unit cell as given by Arachchige et al. 2022 [6]. We provide these patterns in Fig. 8 in the Supplementary information. The DOS pattern of the in-plane orbitals can be seen as three, by a factor of $\sqrt{3}$ enlarged and intersecting Kagome lattices (the same holds for the $d_{xz} + d_{yz}$ orbitals as well as for the EFGs shown in Fig. 3 f). Contrastingly, since the d_{z^2} orbitals of V2 and V3 have the same DOS, they form hexagons centered inside the enlarged Kagome pattern of V1. This configuration resembles the one shown by Arachchige et al. 2022 [6]." as well as a new section and figure in the Supplementary information (Fig. 8).
13. We replaced "100 ppm" by "10 kHz (~ 100 ppm or ~ 1 mT at 8.73 T external field). This implies, for example, that a magnetic moment of $< 10^{-2} \mu_{\text{B}}$ localized at the center of V hexagons (1.75 Å distance to V [6]) remains undetectable with our setup, while, perhaps, be sensed by muon-spin rotation [25]." at the beginning of the discussion.
14. We replaced "of the Kagome lattice is conserved" by "from the high T experiments is retained (cf. Supplementary Fig. 5). The sixfold rotational symmetry reflects the threefold symmetry of a Kagome layer in the CDW regime and that the two neighboring planes of the Kagome double layers are mirror images of each other." in the caption of Fig. 3.
15. We added "It should be noted, however, that the reported CDW phase chiefly considers displacements of Sn along the c -direction, which may have only a very weak effect on the V NMR. Possible consequences are unusual EFG components (splitting or broadening) or an unusual contribution to the spin-lattice relaxation due to the corresponding fluctuations that vanish during the CDW phase transition [12,14]. Non of these signatures were observed." in the discussion.
16. We moved the subsection "CDW phase transition and fluctuations of the local magnetic field" before "Local properties of the CDW phase".
17. We added "Finally, our observation may also be related to the experimental results of a significant loss in carrier concentration during cooling, on the basis of which a pseudogap behavior has been proposed [49]." including the reference at the end of the discussion.

18. We added "about" in the abstract, "about" and "(1.42(2))" in the introduction, added "about" in the caption of Fig. 5, added "(1.95)", replaced " $\sqrt{2}$, in agreement with" by "1.44, i.e., very close to $\sim \sqrt{2}$ as expected from" in the last paragraph of the results section, and we added "about" twice, at the end of the discussion and in the conclusions.
19. We labeled more V atoms by V1, V2, V3 and included the corresponding unit cell in Fig. 2.
20. We included the unit cell in Fig. 3f.
21. We added "A quantitative assessment of the scenario discussed above requires a detailed knowledge of the corresponding hyperfine interaction, which are notoriously difficult to calculate. That is, although typically isotropic, it cannot be excluded that contact interaction or core polarization contribute to the NMR shift and the observed changes, similarly the orbital shift component. From spin-lattice relaxation that most potentially reflects the electronic spin susceptibility only (not orbital), on the other hand side, we calculate 600 before (~ 100 K) and about 400 ppm after (~ 80 K) the phase transition based on the Korringa relation, $K^2 T_1 T = (\gamma_e/\gamma_n)^2 (\hbar/4\pi k_B)$ [50]. The ~ 200 ppm implied changes from T_1 fit quite well the drop of the average shift (gray circles in Figs. 4 **c** and **d**) during the phase transition, as well as the observed splittings. This, however, seems not exactly be reflected in the orbital resolved DOS, where the overall drop due to the phase transition is larger than the splitting, pointing at a more complex shift phenomenology. Similarly, the temperature dependence observed in the relaxation at higher temperatures (800 to 600 ppm between 300 and 100 K) implies a temperature dependent NMR shift which is only partly observed for $c \parallel B_0$ (about 100 ppm)." at the end of the discussion.

Changes due to formatting

In the following, we list the changes due to formatting requirements.

1. We carefully shortened the abstract..
2. We re-arranged the introduction and results section (moved to subheadings from introduction to results).
3. We changed the title of Fig. 1 to "Commensurate charge density waves and nuclear magnetic resonance".
4. We changed the first subheading in results to "Charge density waves and nuclear magnetic resonance".
5. We combined the subheadings "Structural phases of ScV_6Sn_6 " and "Local Vanadium charge symmetry" in the results.
6. We defined abbreviations in captions where needed.
7. We added the title "Electric field gradients at Vanadium sites" to Table 1.
8. We replaced "CDW" by "Charge density wave" in the title of caption Fig. 4 and 5.
9. We changed the reference style to supplementary figures according to formatting instruction several times.
10. We changed math expressions from " x/y " to " $\frac{x}{y}$ " or " xy^{-1} ".
11. We corrected and updated references where needed.

REVIEWERS' COMMENTS

Reviewer #1 (Remarks to the Author):

The authors have already addressed my questions and comments appropriately. Now, I recommend that this paper be published in NC.

Reviewer #2 (Remarks to the Author):

The authors have made considerable efforts to address my concerns in the response and revised manuscript, I recommend the publication of this work. There are a few additional issues that authors should also consider:

The proposed orbital selective CDW modulation is certainly interesting, but it seems challenging to further test it experimentally. An assumption is that the modulation is described by a single sinusoidal. I think it is possible that the realistic modulation cannot be captured by a single sinusoidal function. It is unclear how robust the conclusions are if one allows for more complicated models. Can the authors think of potential tests for this proposed model? Would this model lead to other unusual properties?

Figs. 6c and 6d, make the in-plane plots with the kagome structure contain more unit cells in the ab plane. Or add some comparison plots for different layers in Fig. 8b and 8c in the SI.

In Figs. 6c and 6d, the colors of the V sites do not match for the ab-plane 1d plots (orange and blue need to be switched in one of the plots).

The statement “Finally, we neither find direct evidence of an unusual magnetism as related to time-reversal symmetry breaking and an AHE, nor of an additional, high temperature charge modulation with $q_2 = (1/3, 1/3, 1/2)$ ” is somewhat misleading. NMR may not see the small moment associated with the TRSB (but places an upper limit on the moment, unfortunately not a very small one); q_2 CDW is detectable by single crystal XRD [PRM 7, 104201 (2023)], there is no doubt about its presence, the question is then why V NMR does not see it. Maybe revise to something along the line of “Finally, our NMR measurement do not reveal an unusual magnetism as related to time-reversal symmetry breaking and an AHE, nor of an additional, high temperature charge modulation with $q_2 = (1/3, 1/3, 1/2)$, placing an upper limit of ... on any magnetic moment and suggests the short-range q_2 CDW is due to”

Reviewer #3 (Remarks to the Author):

The authors have addressed the concerns of both referees to satisfaction. I recommend that it be published at this time.

Detailed response to the reviewers

In the following a detailed point-by-point response to the review reports is provided. Any corresponding changes to the manuscript are summarized in the **List of changes** (LOC).

Reviewer #1 (Remarks to the Author):

R1: The authors have already addressed my questions and comments appropriately. Now, I recommend that this paper be published in NC.

A: Thank you very much.

Reviewer #2 (Remarks to the Author):

R2: The authors have made considerable efforts to address my concerns in the response and revised manuscript, I recommend the publication of this work.

A: Thank you very much. We highly appreciate the referee's effort to review our work.

R2: There are a few additional issues that authors should also consider:

R2: The proposed orbital selective CDW modulation is certainly interesting, but it seems challenging to further test it experimentally. An assumption is that the modulation is described by a single sinusoidal. I think it is possible that the realistic modulation cannot be captured by a single sinusoidal function. It is unclear how robust the conclusions are if one allows for more complicated models. Can the authors think of potential tests for this proposed model? Would this model lead to other unusual properties?

A: As mentioned in the discussion, a calculation of the various hyperfine interactions (contact, core polarization, dipole, and orbital) would be an important first step to test our model. Especially the orbital contribution is very large and should thus, partially, also show up in susceptibility measurements. Here, one would need to separate spin from orbital contributions of the electronic susceptibility. Contact interaction, on the other hand, should be rather small, because of the vanishing vanadium s-like character at the Fermi level. Core polarization should also be comparatively small for vanadium. A theoretical treatment would allow to separate all these contributions and to reveal their properties as, e.g., if they contribute to the isotropic or the anisotropic shift. From the experimental point of view, investigating similar V kagome systems with CDWs is a natural next step. Although this cannot prove or disprove our model, the perspective of orbital selective modulations has, to our knowledge, not yet been taken into account.

R2: Figs. 6c and 6d, make the in-plane plots with the kagome structure contain more unit cells in the ab plane. Or add some comparison plots for different layers in Fig. 8b and 8c in the SI.

A: Thanks for this suggestion. We added another panel in Supplementary Fig. 8 which provides the full CDW phase unit cell and the corresponding DOS modulations for the in-plane and the out-of-plane orbitals. Point 1 in LOC.

R2: In Figs. 6c and 6d, the colors of the V sites do not match for the ab-plane 1d plots (orange and blue need to be switched in one of the plots).

A: Thank you very much for pointing this out. We corrected the colors. Point 2 in LOC.

R2: The statement "Finally, we neither find direct evidence of an unusual magnetism as related to time-reversal symmetry breaking and an AHE, nor of an additional, high temperature charge modulation with $q_2 = (1/3, 1/3, 1/2)$ " is somewhat misleading. NMR may not see the small moment associated with the TRSB (but places an upper limit on the moment, unfortunately not a very small one); q_2 CDW is detectable by single crystal XRD [PRM 7, 104201 (2023)], there is no doubt about its presence, the question is then why V NMR does not see it. Maybe revise to something along the line of "Finally, our NMR measurement do not reveal an unusual magnetism as related to time-reversal symmetry breaking and an AHE, nor of an additional, high

temperature charge modulation with $q_2 = (1/3, 1/3, 1/2)$, placing an upper limit of . . . on any magnetic moment and suggests the short-range q_2 CDW is due to”

A: The upper limit of a magnetic moment that we provide is a rather pessimistic estimation of a single moment at the center of the hexagonal voids interacting with V nuclei solely via dipole-dipole couplings. One could also take another point of view. Typically, the NMR line broadening in single crystals is given by the dipole-dipole interactions of the neighboring nuclear moments, i.e., 4 V nuclei and 2 Sn nuclei. Thus, any additional magnetic moment cannot be much larger than these nuclear moments. We think however, since the phenomenon of TRS breaking remains elusive, to take a rather conservative stance. Concerning the additional q_2 CDW modulation, providing an upper limit is more difficult as in the case of a magnetic moment, because a CDW may show up in the local magnetic field at the nucleus as a shift (i.e., DOS), but it may also show up in the local electric field and affects the electric quadrupolar interaction. The quadrupole interaction (QI) can be extremely sensitive, may reflect Fermi level states, bound states as well as ionic neighbors. Furthermore, any slight change typically leads to significant broadening of the satellites, because the variation multiplies with the order of the satellite (progressive broadening from inside to the outside). Above the transition temperature of ~ 96 K, the quadrupole pattern is essentially temperature independent and the satellites have the same width as the central line (the latter is only affected by the dipole-dipole broadening mentioned above). So, there is no room for any variation of the local vanadium environments within our experimental resolution, they are all equivalent in terms of the local electric field gradient. We think the statements in our manuscript leave enough room for present and future experiments to provide evidence for TRS breaking as well as high temperature short range CDW modulation. Point 3 in LOC.

Reviewer #3 (Remarks to the Author):

R3: The authors have addressed the concerns of both referees to satisfaction. I recommend that it be published at this time.

A: Thank you very much.

List of changes

1. We added panel (d) to Supplementary figure 8.
2. We exchanged the orange and blue balls in panel (d) of Fig. 6.
3. We added "with NMR" in the last paragraph of the introduction.